# Structure–Property Relationship in Melt-Spun Poly(hydroxybutyrate-co-3-hexanoate) Monofilaments

**DOI:** 10.3390/polym14010200

**Published:** 2022-01-04

**Authors:** Figen Selli, Rudolf Hufenus, Ali Gooneie, Umit Halis Erdoğan, Edith Perret

**Affiliations:** 1Department of Textile Engineering, Dokuz Eylul University, Izmir 35397, Turkey; figen.selli@ogr.deu.edu.tr (F.S.); umit.erdogan@deu.edu.tr (U.H.E.); 2Laboratory for Advanced Fibers, Empa, Swiss Federal Laboratories for Materials Science and Technology, Lerchenfeldstrasse 5, 9014 St. Gallen, Switzerland; rudolf.hufenus@empa.ch (R.H.); ali.gooneie@empa.ch (A.G.); 3Center for X-ray Analytics, Empa, Swiss Federal Laboratories for Materials Science and Technology, Überlandstrasse 129, 8600 Dübendorf, Switzerland

**Keywords:** poly(hydroxybutyrate-co-3-hexanoate) (PHBH), melt-spinning, structure-property relationship

## Abstract

Poly(hydroxybutyrate-co-3-hexanoate) (PHBH) is a biodegradable thermoplastic polyester with the potential to be used in textile and medical applications. We have aimed at developing an upscalable melt-spinning method to produce fine biodegradable PHBH filaments without the use of an ice water bath or offline drawing techniques. We have evaluated the effect of different polymer grades (mol% 3-hydroxy hexanoate, molecular weight etc.) and production parameters on the tensile properties of melt-spun filaments. PHBH monofilaments (diameter < 130 µm) have been successfully melt-spun and online drawn from three different polymer grades. We report thermal and rheological properties of the polymer grades as well as morphological, thermal, mechanical, and structural properties of the melt-spun filaments thereof. Tensile strengths up to 291 MPa have been achieved. Differences in tensile performance have been correlated to structural differences with wide-angle X-ray diffraction and small-angle X-ray scattering. The measurements obtained have revealed that a synergetic interaction of a highly oriented non-crystalline mesophase with highly oriented α-crystals leads to increased tensile strength. Additionally, the effect of aging on the structure and tensile performance has been investigated.

## 1. Introduction

Polyhydroxyalkanoates (PHAs) have attracted great interest because of their processability, biocompatibility, and biodegradability. PHAs are biologically derived thermoplastic polyesters produced by various microorganisms [1,2]. Poly(3-hydroxybutyrate-co-3-hydroxyhexanoate) is typically abbreviated as P(3HB-co-3HH) or simply PHBH and is produced by microorganisms under fermentation conditions using plant oils. This environment-friendly polymer shows biodegradability under aerobic, anaerobic, aquatic, and compost conditions without any toxic residual [3,4]. PHBH has a lower melting point (T_m_ = 125–150 °C) than poly(3-hydroxybutyrate) (P3HB) (T_m_ = 175–180 °C) and poly(3-hydroxybutyrate-co-valerate) (PHBV) (T_m_ = 170–175 °C) [5,6]. Compared to the P3HB homopolymer, PHBH reveals a faster degradation rate, an improved toughness and heat resistance, as well as a broader thermal processing window [7].

PHBH is a random copolymer that consists of 3-hydroxybutyrate (3-HB) and 3-hydroxyhexanoate (3-HH) segments (Figure 1). The 3HH medium-length chains connect the main 3HB chains.

The mole percentage of the 3-HH unit has a prominent effect on thermal properties, such as melting temperature or heat of fusion [8,9]. The structural properties of PHBH, like its crystallization behavior, and the effect of aging on the mechanical properties of PHBH, containing different amounts of 3-HH, have been investigated in the past [10,11,12,13]. A high mole percentage of the 3-HH unit leads to lower melting temperatures, lower crystallinities, and reduced storage modulus and overall strength. On the other hand, PHBH copolymers with low 3-HH contents undergo physical aging over time, leading to an increase in modulus and strength, but a reduction in ductile properties [10,11]. This latter behavior is explained by a secondary crystallization that occurs above the glass transition temperature, T_g_ [11,14,15].

Numerous thermoplastic processing techniques (e.g., extrusion, injection moulding, melt-spinning) have been applied in the past, and the polymer PHBH was formed into films, coatings, and fibers [16,17,18,19]. The melt-spinning technique is the most cost-effective method to produce fibers, due to a simple, solvent-free production line and a high production rate [20]. PHBH has a relatively low glass transition temperature (about 0 °C) as well as a relatively low melting point (about 145 °C). Melt spinning of PHBH is quite challenging due to its slow crystallization rate and low thermal stability [12,21]. There are only a few studies that discuss the interplay between the structural and mechanical properties of PHBH fibers melt-spun at the laboratory scale. These articles mainly focus on enhancing tensile properties by making specific modifications to the spinning line [19,21,22,23,24]. Most authors have quenched PHBH into an ice water bath to hinder secondary crystallization, followed by various post-drawing techniques [19,22,24]. The thus produced fibers had rather large diameters (several 100 µm), and the reported tensile strengths were mostly rather low (<100 MPa). Kabe et al. [22] could achieve fibers with higher tensile strengths up to 552 MPa using a two-step drawing technique with intermediate annealing conditions, starting from PHBH with 5.5 mol% of the 3HH unit. Wide-angle X-ray diffraction (WAXD) measurements have revealed that the PHBH fibers with high tensile strength had a high contribution of mesophase, which gave rise to a broad reflection on the equator in WAXD patterns [7,25]. This mesophase also exists in highly drawn P3HB fibers [26,27,28,29], and its exact structure has been a matter of debate. Many authors have called this phase the β-form phase, where the chains are thought to have a zigzag conformation. A crystalline structure should however give rise to many additional off-axis reflections in the WAXD patterns, which in reality are missing. We therefore believe that the mesophase in PHBH fibers is, as in other fibers [26], made of highly oriented but conformationally disordered (tie-)chains between α-form crystals. Thus, in analogy to the mesophase in P3HB, we call this phase a highly oriented non-crystalline phase, P_nc_. Qin et al. [21] have produced PHBH filaments with a high-speed (take-up speed of 6 km/min) melt-spinning method, where the filaments reached tensile strengths of up to 156 MPa. In this study, the crystalline orientation of α-form crystals increased with an increase in the take-up godet speed and the existence of a small amount of mesophase was detected at high take-up speeds. The mechanical properties of the PHBH fibers could be improved (tensile strength of 215 MPa) by installing a liquid isothermal bath (water-soluble polypropylene glycol) in the high-speed spinning line [23].

Our presented melt-spinning and online drawing method provides a guideline for economic industrial-scale (upscalable) production without the use of an ice water bath and post-drawing techniques. We have melt-spun fine PHBH monofilaments (diameter < 130 µm) from three different polymer grades (3-HH of 6 and 11 mol%) using a pilot-scale melt-spinning plant and have studied the influence of polymer grades and processing parameters on tensile properties. Thermal properties and rheological properties of the different polymer grades have been investigated, and morphological, thermal, mechanical, and structural properties of the melt-spun filaments thereof are discussed. Differences in tensile properties have been correlated with structural differences between the fibers using wide-angle X-ray diffraction (WAXD) and small-angle X-ray scattering (SAXS). Additionally, the effect of aging (storage for 33 months) on the tensile properties, as well as on the fiber structure, has been investigated.

## 2. Materials and Methods

### 2.1. Materials

PHBH monofilaments were melt-spun from three different polymer grades provided by Kaneka Corporation (Osaka, Japan) in the form of pellets. Physical properties of the individual polymer grades are given in Table 1. All polymer grades include both slipping and nucleating agents.

The copolymers have either 11 mol% (grade I) or 6 mol% (grade II, III) of 3-HH. The polymer grade I has a higher 3-HH content and lower melting temperature (T_m_ = 126 °C) than the other grades. The absolute average molecular weights are higher in grades I and III. The dispersity [30] (colloquially called “polydispersity”) is however equal for all polymer grades, which indicates a similar distribution of molecular weights in all grades. Here, grade II is intended for, e.g., injection molding, but usually not for melt-spinning due to its low melt-strength. In melt spinning, a sufficiently high molecular weight is required in order to prevent filament breakage in the spin-line.

### 2.2. Melt-Spinning of PHBH Monofilaments

Melt-spinning of PHBH monofilaments was carried out on a custom-made pilot-scale melt-spinning plant (Fourné Polymertechnik, Germany) described elsewhere [31]. Prior to melt-spinning, the PHBH polymers were dried in a vacuum oven at 90 °C for 12 h. Drying of polymers is very important before extrusion since moisture can strongly influence processability. Polyesters can suffer considerable loss in molecular weight by hydrolytic degradation (hydrolysis) of the melt in presence of water [20]. PHBH pellets were melted and pressurized using an 18 mm screw extruder with a length to diameter (L/D) ratio of 25. To build up the required spin pressure, and to provide a controlled throughput of the melt, a gear pump with a capacity of 0.6 cm^3^ per revolution was used. A summary of the production parameters is given in Table 2. The first number in the fiber labels, e.g., 1780, corresponds to the fiber number assigned by Empa, where increasing numbers signify a later date of production. The second number, e.g., I, signifies the used polymer grade. The three heating zones of the extruder were set to temperatures between 140 and 160 °C. The melt temperatures measured in the spin pack ranged between 150 and 166 °C, and the circular monofilament die had a diameter of 0.5 mm and a capillary length to diameter (L/D) ratio of 4. The polymer melt leaving the spin pack was cooled down in an air cooling chamber with a height of 1.2 m. Six or seven godets were used to draw the fibers online, and the godet temperatures varied between 10 and 70 °C. The draw ratio (DR) was calculated as the ratio of winding speed and take-up speed of the first godet. In total, nine monofilaments from three different polymer grades were melt-spun. Note that fibers were successfully melt-spun even from the injection moulding polymer grade II.

### 2.3. Analysis of Thermal Properties

The melting and crystallization behaviors of the PHBH pellets and filaments were characterized with differential scanning calorimetry (DSC) using a calorimeter from TA instruments (DSC Q20, TA, New Castle, DE, USA) with a temperature accuracy of ±0.1 °C. Measurements were performed in a nitrogen atmosphere. The following heating and cooling cycles were applied: first heating from 25 °C to 200 °C, followed by a cooling step to −25 °C and a second heating back to 200 °C. The heating and cooling rates were set to 10 °C/min. The crystallinity was estimated from the determined melt enthalpy, ΔH_melt_ of the first heating cycle for the filaments, and of the second heating cycle for the PHBH pellets, according to the following equation:(1)$$\%\;crystallinity=\frac{{\Delta H}_{\mathrm{melt}}}{{\Delta H}_{\mathrm{ref}}}\times 100$$
where the 100% crystalline PHBH melting enthalpy, ΔH_ref_, is 146 J/g [11,32].

The thermal stability of different polymer grades was analyzed with thermogravimetric analysis (TGA). TGA (Shimadzu, DTG-60H, Japan, accuracy ± 1 °C) was performed under nitrogen increasing the temperature from 25 °C to 700 °C. The heating rate was set to 10 °C/min.

### 2.4. Analysis of Rheological Properties

The flow behaviour of different PHBH grades was studied at 165, 170, and 190 °C with a rotational rheometer (Anton Paar, Physica MCR 301, Graz, Austria), using a plate-plate geometry, with a plate diameter of 25 mm and a 1 mm gap. Viscosity, storage, and loss modulus were examined at a constant strain amplitude and angular frequency of 1% and 1 rad/s, respectively, as a function of time in order to investigate the thermal degradation behavior of the different PHBH polymer grades.

### 2.5. Analysis of Morphological Properties

Morphological properties of melt-spun PHBH filaments were analysed using a scanning electron microscope (Carl Zeiss 300VP, Germany) with an acceleration voltage of 3–5 kV. The longitudinal fiber samples were coated with 8 nm gold prior to SEM observation.

### 2.6. Analysis of Fiber Diameters

An optical microscope (Olympus BX43, Tokyo, Japan) was used to analyze the diameter of PHBH monofilaments by averaging over 20 measurements. Additionally, we have also calculated the diameter of the fibers from the measured linear density, in tex = mg/m. The linear density is a measure of the mass per unit length of a fiber, also called fineness [33]. A mass density of 1.19 g/cm^3^ (fibers of polymer grade I) and 1.20 g/cm^3^ (fibers of polymer grade II and III) was used for the calculations of the diameters.

### 2.7. Analysis of Mechanical Properties

Mechanical testing of the filaments was performed with the tensile testing machine Zwick/Roell Retroline Z100, Switzerland. The load–strain behaviour was evaluated with a 10 N load cell in reference to the standard ASTM D 2256. Filament tests were performed with a gauge length of 50 mm using a constant rate of extension of 100 mm/min. Tensile strength and elongation at break values were obtained by averaging over 20 measurements. Tensile tests were repeated with filaments aged 33 months at room temperature after the production date.

### 2.8. Analysis of Structural Properties

Wide-angle X-ray diffraction (WAXD) and small-angle X-ray scattering (SAXS) patterns were recorded using a Bruker Nanostar U diffractometer (Bruker AXS, Karlsruhe, Germany). The instrument is equipped with a VÅNTEC-2000 MikroGap area detector and a Cu-Kα radiation (λ = 1.5419 Å) source and a pin hole with a diameter of 300 µm. WAXD and SAXS measurements were performed, for a selection of filaments, a few days after the production date, with an exposure time of 1800 s for WAXD and 3600 s for SAXS, respectively. Additionally, filaments were remeasured with WAXD, 33 months after production and with SAXS, 30 months after production. The latter WAXD patterns were acquired with either 3600 s or 4600 s exposure times and a sample to active detector area distance of sd ~ 94 mm. The latter SAXS patterns were acquired with 4600 s exposure times and sd ~ 111 mm.

## 3. Results and Discussion

### 3.1. Thermal and Rheological Properties of PHBH Polymer Grades

Melting and crystallization properties of PHBH polymers were characterized using differential scanning calorimetry (DSC). The first heating cycle was conducted to erase the thermal history of the polymer grades. The thermograms of the 2nd heating and 1st cooling cycles are presented in Figure 2.

Thermal properties of PHBH polymer grades obtained from DSC curves, including the melting temperature (T_m_), melting enthalpy (ΔH_m_), crystallinity (X_c_), crystallization temperature (T_c_), and crystallization enthalpy (ΔH_c_), are summarized in Table 3. The temperatures T_m_ and T_c_ correspond to the locations of the maxima of the melting peaks (2nd heating) and crystallization peaks (1st cooling), respectively.

The second heating curves in Figure 2a show that two sharp melting peaks exist for polymer grades II and III. One melting peak is located at a lower temperature of around 133 °C and one at a higher temperature of around 146 °C. The existence of two melting peaks for semi-crystalline polymers has been widely reported [34,35] and a number of hypotheses that explain such a melting behavior have been proposed. The most common mechanism of this multiple melting behavior is the melting–recrystallization–remelting that depends on polymer structure and crystallization conditions [36,37,38]. Hu et.al [39] studied the multiple melting behavior of PHBH polymer (3HH of 12 mol%) during the heating process and concluded that the first endothermic peak corresponds to the melting of crystals formed during the primary crystallization, and that the second peak can be ascribed to the melting peak of crystals formed by recrystallization during the heating process. Overall, grade II and grade III show very similar DSC curves, although the crystallization peak is located at a slightly higher temperature. In contrast, the first melting peak in polymer grade I is very weak and broad (starts from ~100 °C), and the crystallization peak is also broad and weak. Thus, grade I is less crystallizable and forms less perfect crystals during the cooling cycle.

The thermal degradation of PHBH polymer grades was investigated by thermogravimetric analysis (TGA). The thermal stability of polymers during extrusion is crucial to enhance the processability and to prevent degradation. The TGA curves of PHBH polymer grades show a single-step thermal degradation process. The results show that all the PHBH polymer grades start to degrade above 275 °C (Figure 3), which is about 100 °C higher than fiber extrusion temperatures. The temperature of maximum degradation rate (T_max_), which indicates the point of greatest rate of weight change, was similar for all grades (Figure 3). The weight loss of the polymers at T_max_ was determined to be 63.4%, 52.7%, and 63.8% for grades I, II, and III, respectively.

Rheological properties (viscosity, loss, and storage modulus) have been analyzed with a rheometer as described in the experimental section. Figure 4 shows the dependence of the complex viscosities of the different grades at three temperatures, 165 °C, 170 °C and 190 °C. Polymer grade II has the lowest viscosity at the beginning of the experiments due to its lower molecular weight of 300 kDa and shows a gradual decay over time. Polymer grade III, with a higher molecular weight of 500 kDa, has the highest initial viscosity of ~14,300 Pa·s (Figure 4). With a temperature increase from 165 °C to 190 °C, all polymer grades exhibit a decrease in complex viscosity.

Figure 5 shows the loss (G″) and storage modulus (G′) of different PHBH grades versus time at different testing temperatures. All polymer grades show rapid degradation pathways in their molten states at all temperatures. The decay in melt rheological properties becomes stronger as the temperature is increased to the maximum temperature (190 °C). Similar to the complex viscosity results, polymer grade III has the highest initial moduli at all of the temperatures. Following these findings, it was decided to keep the extrusion temperatures ≤ 160 °C to minimize degradation during melt-spinning of fibers.

### 3.2. Morphological Properties and Fineness of PHBH Monofilaments

Nine different monofilaments (diameter < 130 μm) were melt-spun from PHBH grade I (1780, 1781, 1782), grade II (1848, 1783, 1784), and grade III (1845, 1846, 1847), respectively. Production parameters, such as draw ratio and take-up speed, can have a significant effect on the fiber morphology [19], which is in direct relation with end-use product characteristics like optical properties and tactile comfort. Scanning electron microscopy (SEM) was used to study the surface topography of the filaments (Figure 6). Almost all fibers show smooth surfaces. Only fiber 1847-III shows fibril-like lines running parallel to the drawing direction. This particular fiber was drawn with a lower take-up godet temperature and a lower throughput rate, which may have increased the friction (stickiness) between take-up godet and fiber.

As mentioned in Section 2.6, the diameters of the individual fibers have been determined from optical micrographs (observed) as well as from the measured fineness (calculated). The corresponding values are summarized in Table 4.

### 3.3. Thermal Properties of PHBH Monofilaments

The monofilaments from polymer grade I (Figure 7a) have broader melting peaks and crystallization peaks (first heating and cooling curves), similar to the DSC curves of the polymer (Figure 2), which indicates that these filaments have a lower crystallinity and a less perfect crystalline structure. In contrast, sharp melting and crystallization peaks are observed for the PHBH monofilaments that were produced from polymer grade II and III (Figure 7b,c).

Thermal properties of PHBH monofilaments derived from the DSC curves are given in Table 5. The monofilaments have melting and crystallization temperatures that correspond to the respective polymer grades (Figure 2). Besides, the heating curves of the monofilaments show that there are also two sharp melting peaks, which is similar to the melting behavior of the initial polymer (Section 3.1).

During online drawing, a first crystallization most-likely happens during quenching of the melt after exiting the die, before the filament hits the first godet. Further crystallization occurs during the drawing of the filament (strain-induced crystallization). Thus, the monofilaments 1781-I, 1782-I (DR = 10) from polymer grade I have higher melting enthalpies compared to 1780-I (DR = 6.5).

The calculated crystallinity for the filaments melt-spun from polymer grade I was rather low (<27%). In contrast, filaments melt-spun from grade II and III revealed crystallinities of over 32%. Molecular weight and molecular weight distribution of polymers have a prominent effect on the crystallization behavior and play an important role during spinning [40,41]. From Table 3, it can be seen that fibers from polymer grade III, having a higher molecular weight (500 kDa) than polymer grade II (300 kDa), had slightly higher crystallinity values (>34%) than fibers produced from grade II.

For filaments melt-spun from grade III, an increase in the crystallization temperature (T_c_) was noticed. This suggests that, in the first cooling cycle, crystals already start to grow at higher temperatures. A similar phenomenon of increasing T_c_ with increasing molecular weight has been reported for ultra-high molecular weight PHBH [42].

### 3.4. Mechanical Properties of PHBH Monofilaments

Table 6 lists mechanical properties of freshly spun and aged (33 months) PHBH monofilaments, as a function of draw ratio. Tensile strength and elongation at break range from 28 to 291 MPa and 15–263%, respectively. Tensile properties of melt-spun fibers depend on the molecular orientation, which in turn is mostly determined by the draw ratio of the fiber [40]. It is known that the crystallization rate of PHBH is slow, and that the polymer is in a rubbery state at room temperature due to its low T_g_. Drawing can increase the rate of crystallization during fiber spinning [43,44]. Accordingly, it was found that the tensile strength was higher for a higher draw ratio in the case of fibers melt-spun from polymer grade I (1780, 1781, 1782) and polymer grade III (1845, 1846, 1847) (Table 6). The filaments 1784-II and 1846-III exhibit the highest tensile strengths of 206 MPa and 291 MPa, respectively. Melt-spun filaments of polymer grade I, with higher 3HH content (11 mol%) and lower crystallization temperature (50–60 °C), exhibit the lowest tensile strength among all fibers.

Filament 1784-II has a higher tensile strength than 1783-II and 1848-II. Comparing the production parameters of these fibers (Table 2), 1784-II was produced with a relative low take-up speed (100 m/min), possibly promoting solidification before take-up and thus increasing tensile stress and molecular orientation in the spin-line. The structural analysis in Section 3.5 shows how these slightly different production parameters affect the structure and thus enhance the tensile properties.

According to the tensile test results, the elongation at break decreased for increasing draw ratio, due to the increased alignment of molecular chains along the drawing direction. The highest elongation at break value (263.1%) was measured for the filament 1780-I with the lowest draw ratio of 6.5.

As mentioned in the introduction section, PHA polymers undergo physical aging, which is ascribed to secondary crystallization, i.e., to the rearrangement of molecules into energetically more favorable structures [14,43,45]. This secondary crystallization reduces the number of polymer chains in the amorphous region, resulting in increased brittleness of the fibers. The tensile tests of aged (33 months storage at room temperature) filaments showed that the tensile strength has increased for almost all filaments, and that the elongation at break decreased (Figure 8, Table 6). This suggests that the crystallinity has indeed increased over time. Only for fibers (1784-II, 1846-III) that had already a relatively high ultimate tensile strength (UTS) directly after melt-spinning is a decrease in UTS upon aging observed. For these filaments, the tensile potential was already exhausted during the spin-drawing, and sustained stress during aging most-likely leads to the local overstretching and breakage of macromolecules.

### 3.5. Structural Properties of PHBH Monofilaments

We have analyzed the structural properties of aged PHBH filaments with wide-angle X-ray diffraction and small-angle X-ray scattering. SAXS patterns are shown in the Appendix A, where the extraction of long-spacing is also explained. The long-spacing, i.e., the average distance between highly oriented α-crystals, has been found to be close to 6 nm for all fibers. Measured 2D WAXD patterns are shown in Figure 9. The positions of the reflections (except one additional equatorial reflection) are conform with an orthorhombic, P2_1_2_1_2_1_ (D2^4^) unit cell (α = β = γ = 90°) [7,25], similar to the one in α-crystals of the P3HB homopolymer [46,47,48]. Monofilaments melt-spun from polymer grade I show arc-shaped reflections (Figure 9a–c), whereas the fibers melt-spun from grades II and III show sharp reflections. Arc-shaped reflections signify that the α-crystals in fibers melt-spun from grade I are less oriented. Interestingly, in analogy to drawn P3HB fibers, an additional broad equatorial reflection appears in highly drawn PHBH fibers, which cannot be attributed to the α-phase, but to a highly oriented non-crystalline mesophase instead. As mentioned in Section 1, this mesophase used to be called “β-phase”. We have, however, no indication that the mesophase is crystalline and that the chains adopt a planar zigzag conformation (β-form). Thus, we prefer to call this phase a highly oriented non-crystallline mesophase, P_nc_, which we have previously introduced for P3HB [27,28,29]. Such a mesophase is typically located in-between α-crystals in semi-crystalline polymers and is made of highly-oriented tie molecules [26]. The mesophase can, however, also develop from the amorphous phase. The highly oriented (along the fiber axis), but conformationally disordered chains, give rise to broad equatorial reflections in WAXD patterns (Figure 9f), where the position of the reflections reflect the average lateral spacings between the molecular chains. Such mesophases have been observed in the past for fibers melt-spun from the homopolymer P3HB [27,28,29], or for fibers melt-spun from other semi-crystalline polymers (polyethylene terephthalate [49], polyethylene, polypropylene, polyamide 6 etc.) or amorphous polymers (polycarbonate, cyclo-olefin polymer, copolyamide, polyethylene terephthalate glycol) [26]. This mesophase has been shown to strongly affect the mechanical properties in P3HB or poly-ε-caprolactone (PCL) fibers [50,51] amongst others [20].

We have quantified the crystal orientation in the fibers by fitting azimuthal profiles to Pearson VII functions and by calculating the Hermans’ orientation factor [52,53]. Three azimuthal profiles across the equatorial reflections, α(020), α(110), and mesophase, P_nc_, have been analyzed. The corresponding annuli are indicated in Figure 9f with dashed white lines, and range from 2θ = 12–15°, 15–18°, and 18–24°, respectively.

The azimuthal profiles are shown in Figure 10, and the calculated Hermans’ orientation parameters are shown in Figure 11. Note that Hermans’ orientation parameters, that have been calculated from measured WAXD patterns a few days after fiber production, are shown with faded colors, whereas the values from aged fibers are shown as strong colors. For *f*_(*hk*0)_ = 1, the (*hk*0) planes of the crystals are completely aligned parallel to the fiber axis, whereas for *f*_(*hk*0)_ = 0, the crystals are randomly oriented. The α-crystal orientation is considerably higher for fibers melt-spun from grade II and III, compared to grade I. In fibers with mesophase content, the mesophase orientation is very high. Note that fibers, 1780-I and 1781-I, have no mesophase content. In fiber 1781-I, the α-crystals have lost some of their orientation in the 33 months of aging.

Figure 12 shows the measured tensile strength of the aged fibers, in comparison to the peak heights of the α(020) and P_nc_ reflections in the azimuthal profiles. Note that computations of the peak areas have shown the same trends as the peak heights (not shown). The peak heights/areas of the α(020) reflections are correlated with the crystallinity in the samples, and the height of the mesophase reflection is correlated with the amount of mesophase. Note that the trend in the crystallinity deduced from α(020) peak heights is very similar to the trend seen from DSC measurements (Table 5). The tensile strength of the fibers increases from grade I over grade II to grade III (black dots). From Figure 12, it becomes clear that this increase in tensile strength correlates with an increase in mesophase content, an increase in crystallinity, as well as an increase in crystalline orientation. A synergistic effect between the α-phase and the mesophase seems to improve the tensile properties of the fibers. Table 6 has shown an increase in tensile strength and a decrease in elongation at break upon aging for most of the fibers. A comparison of 2D WAXD patterns measured directly after melt-spinning, with WAXD patterns measured on aged fibers, revealed that the mesophase content and crystallinity generally increased over time (see Appendix A), which confirms the occurrence of a secondary crystallization. The increase in crystallinity and mesophase content is responsible for the increase in tensile strength and decrease in elongation at break.

## 4. Conclusions

Nine PHBH monofilaments have been successfully melt-spun with an online drawing technique without the use of an ice-water bath. The presented melt-spinning of PHBH with air quenching and immediate drawing by a set of heated godets is upscalable for economic, industrial-scale production. We have investigated the effect of different PHBH polymer grades (3HH: 6–11 mol%; molecular weight: 300–500 kDa) and production parameters on the mechanical properties of melt-spun, online drawn PHBH fibers. The structure–property relationship was analyzed by performing wide-angle X-ray diffraction and small-angle X-ray scattering experiments. Depending on the polymer properties and production parameters, different mechanical properties were obtained (tensile strength: 28–291 MPa; elongation at break: 15–263%). The highest tensile strengths have been observed for fibers melt-spun from the polymer with highest molecular weight and lowest mol% 3HH. Compared to other biocompatible melt-spun fibers, e.g., poly(ε-carpolactone) (UTS 347 MPa, el. at break 73% [50]) or poly(3-hydroxybutyrate) (UTS 171 MPa, el. at break 24% [43]), PHBH can achieve higher UTS values than P3HB. However, the changes in mechanical properties due to aging are more pronounced in PHBH. Higher UTS can be achieved for PCL fibers, however PCL is less biodegradable than PHBH and is of greater interest for, e.g., long-term implants. The tensile tests of aged PHBH filaments (33 months) showed that for most fibers the tensile strength increased and the elongation at break decreased due to an increase in crystallinity over time. A synergistic interplay between crystallinity, α-crystal orientations, and the amount of mesophase affects the tensile properties.

## Figures and Tables

**Figure 1 polymers-14-00200-f001:**
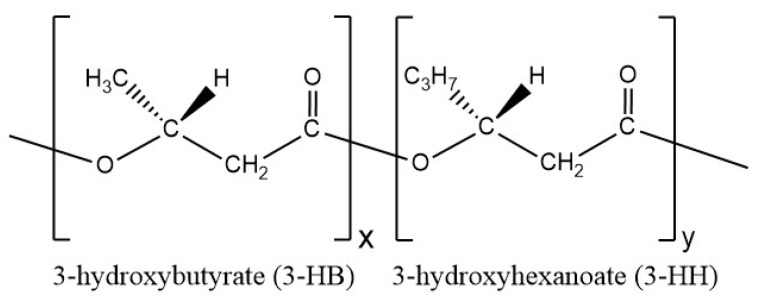
Chemical structure of poly(3-hydroxybutyrate-co-3-hydroxyhexanoate) (PHBH).

**Figure 2 polymers-14-00200-f002:**
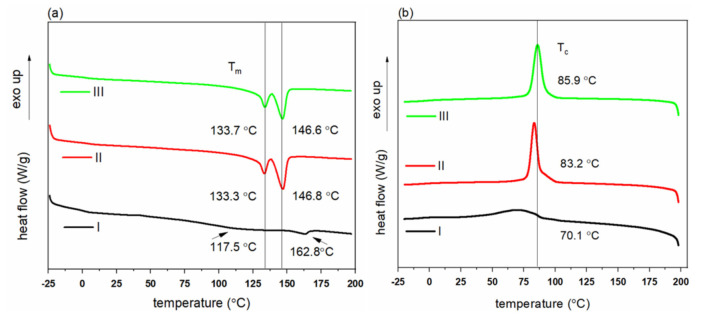
DSC curves of PHBH polymers grades: (**a**) melting curves (second heating), (**b**) cooling curves (first cooling). Curves have been offset vertically for better visibility.

**Figure 3 polymers-14-00200-f003:**
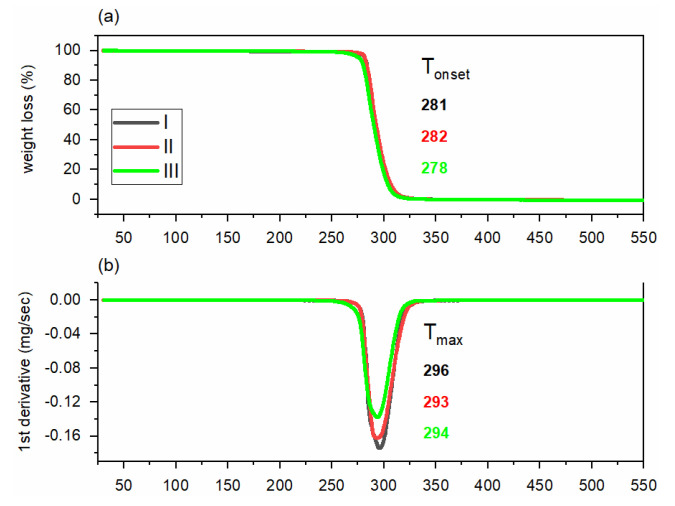
Thermal degradation of PHBH polymer grades; (**a**) TGA degradation curves in terms of mass loss (**b**) first derivative of TGA thermograms.

**Figure 4 polymers-14-00200-f004:**
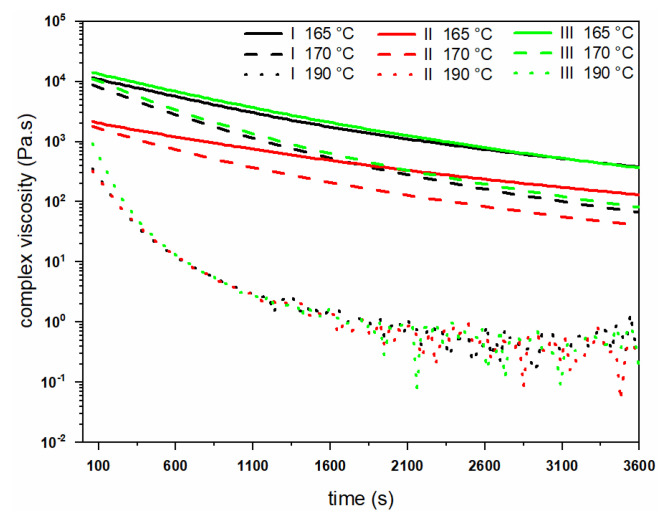
Change in complex viscosity of PHBH polymer grades at 165, 170 and 190 °C.

**Figure 5 polymers-14-00200-f005:**
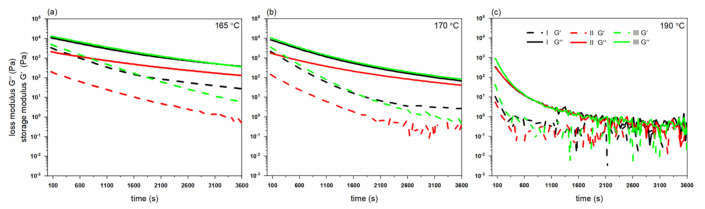
Change in loss (G″) and storage modulus (G′) of PHBH polymer grades at (**a**) 165, (**b**) 170 and (**c**) 190 °C.

**Figure 6 polymers-14-00200-f006:**
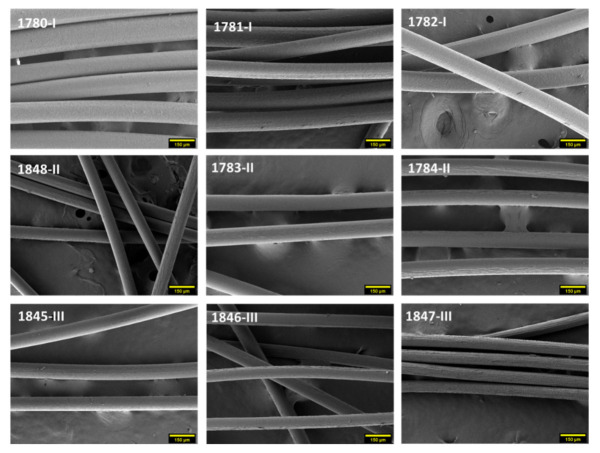
SEM images of PHBH monofilaments (magnification 250×, scale bar 150 μm).

**Figure 7 polymers-14-00200-f007:**
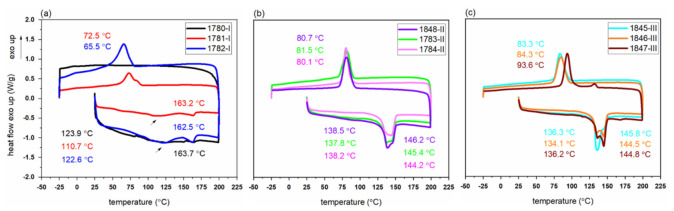
First DSC heating/cooling cycles of PHBH monofilaments produced from polymer grade (**a**) I (**b**) II (**c**) III.

**Figure 8 polymers-14-00200-f008:**
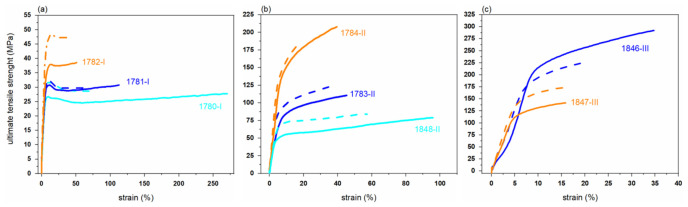
Stress-strain curves of PHBH monofilaments produced from grade (**a**) I, (**b**) II, (**c**) III. Measurements of aged (33 monts) fibers are shown as dashed curves.

**Figure 9 polymers-14-00200-f009:**
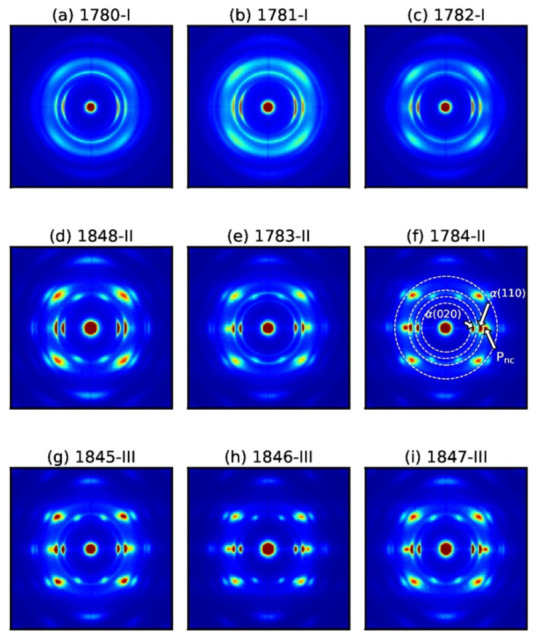
2D WAXD patterns of melt-spun PHBH monofilaments for polymer grade I (**a**–**c**), II (**d**–**f**), III (**g**–**i**). The integrated azimuthal annuli are marked with dashed white lines in panel (**f**).

**Figure 10 polymers-14-00200-f010:**
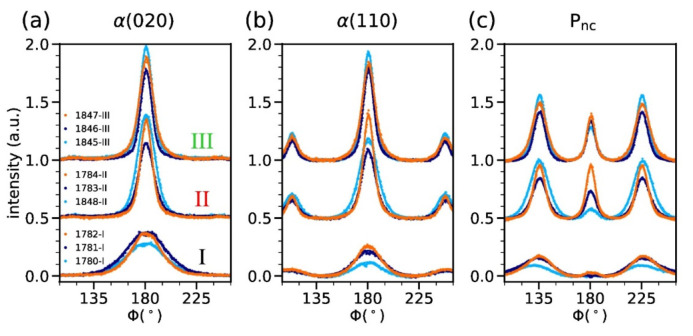
Azimuthal profiles of annulus around the equatorial reflections of (**a**) α(020), (**b**) α(110) and (**c**) P_nc_. The curves of fibers melt-spun from different polymer grades have been offset by 0.5 for better visibility.

**Figure 11 polymers-14-00200-f011:**
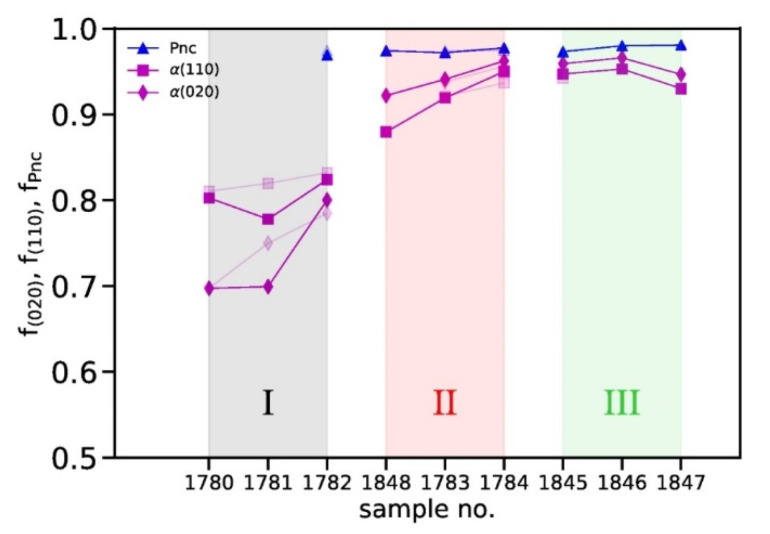
Hermans’ orientation factors as a function of the sample number for 33 months aged fibers. Shaded areas highlight the polymer grades I, II and III. Orientation factors extracted from WAXD patterns measured directly after spinning are shown with a faded color.

**Figure 12 polymers-14-00200-f012:**
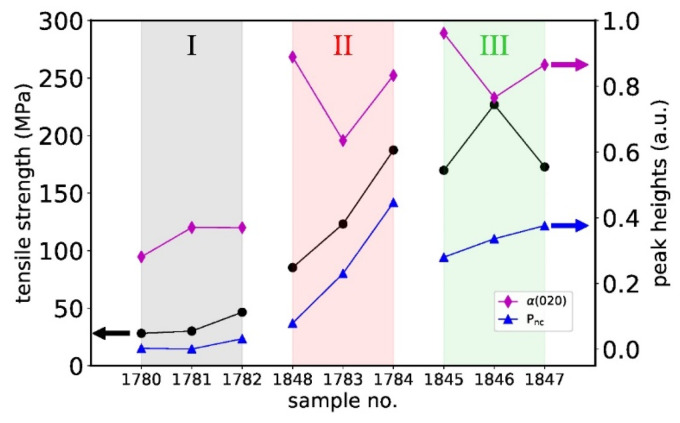
Tensile strength and peak heights of α(020) and P_nc_ reflections in azimuthal profiles as a function of sample number (aged fibers).

**Table 1 polymers-14-00200-t001:** Nominal properties of PHBH polymer grades [3].

Grade	I	II	III
3-hydroxy hexanoate (mol%)	11	6	6
melting point (°C)	126	145	145
glass transition temperature (°C)	0	2	2
crystallization temperature (°C)	50–60	>70–80	70–80
melt flow index (g/10 min) *	3	12	3
molecular weight (kDa)	500	300	500
dispersity [30]	2	2	2
density (g/cm^3^)	1.19	1.20	1.20

* @160 °C, 5 kg.

**Table 2 polymers-14-00200-t002:** Production parameters of PHBH monofilaments.

Fiber No.-Grade	Melt Temp.(°C)	Spin Press. (bar)	Mass Throughput (g/min)	Air Cool. Temp. (°C)	Godet 1 Speed/ Temp. (m/min, °C)	Godet 2 Speed/ Temp. (m/min, °C)	Godet 3 Speed/ Temp. (m/min, °C)	Godet 4 Speed/ Temp. (m/min, °C)	Godet 5 Speed/ Temp. (m/min, °C)	Godet 6 Speed/ Temp. (m/min, °C)	Godet 7 Speed/ Temp. (m/min, °C)	Winder Speed (m/min)	Draw Ratio
1780-I	150	143	5.85	26	60/70	270/50	360/40	-	360/60	375/40	390/30	390	6.5
1781-I	150	150	6.00	20	60/63	300/70	420/60	550/50	550/60	575/50	600/30	600	10.0
1782-I	154	150	7.20	20	60/63	300/65	420/55	550/45	550/60	575/50	600/35	600	10.0
1848-II	153	72	4.00	12.5	100/10	700/60	800/60	800/50	800/40	800/35	800/35	800	8.0
1783-II	153	65	7.68	19	120/50	900/50	950/50	950/40	950/50	955/40	960/30	960	8.0
1784-II	154	68	6.48	19	100/50	-	780/40	805/40	805/40	810/40	810/30	810	8.1
1845-III	157	112	5.95	19	100/30	550/30	600/30	600/40	600/50	600/40	600/30	595	6.0
1846-III	166	90	4.80	16	100/65	700/60	800/60	800/50	800/50	800/40	800/30	800	8.0
1847-III	166	90	4.32	16	100/50	700/60	800/60	800/50	800/50	800/40	800/30	800	8.0

**Table 3 polymers-14-00200-t003:** Thermal properties of PHBH polymer grades.

Grade	Tm (°C)		∆H_m_ (J/g)	X_c_ (%)	T_c_ (°C)	∆H_c_ (J/g)
I	117.5	162.8	15.94	10.92	70.1	27.08
II	133.3	146.8	46.07	31.55	83.2	50.63
III	133.7	146.6	47.38	32.45	85.9	50.42

**Table 4 polymers-14-00200-t004:** Draw ratio, fineness and determined diameters.

Fiber No.-Grade	Draw Ratio (DR)	Fineness (Tex = mg/m)	Diameter (µm) Calculated	Diameter (µm) Observed
1780-I	6.5	15	126.7	119.9 ± 3.9
1781-I	10.0	10	103.5	96.4 ± 4.5
1782-I	10.0	12	113.3	101.1 ± 6.8
1848-II	8.0	5	72.9	74.4 ± 2.1
1783-II	8.0	8	92.2	86.1 ± 1.7
1784-II	8.1	8	92.2	93.9 ± 1.6
1845-III	5.95	10	103.0	96.0 ± 1.4
1846-III	8.0	6	79.8	83.9 ± 3.3
1847-III	8.0	5.4	75.7	78.9 ± 1.3

**Table 5 polymers-14-00200-t005:** Thermal properties of PHBH filaments.

Fiber No.-Grade	T_m_ (°C)		∆H_m_ (J/g)	X_c_ (%)	T_c_ (°C)	∆H_c_ (J/g)
1780-I	123.9	163.7	16.33	11.18	-	-
1781-I	110.7	163.2	17.43	11.94	72.5	29.49
1782-I	122.6	162.5	39.01	26.72	65.5	54.68
1848-II	138.5	146.2	48.24	33.04	80.7	52.30
1783-II	137.8	145.4	48.47	33.20	81.5	43.69
1784-II	138.2	144.2	47.39	32.46	80.1	42.37
1845-III	136.3	145.8	51.78	35.47	83.3	45.05
1846-III	134.1	144.5	52.06	35.66	84.3	47.98
1847-III	136.2	144.8	52.08	35.67	93.6	47.18

**Table 6 polymers-14-00200-t006:** Filament draw ratios, fineness and mechanical properties.

Fiber No.-Grade	Draw Ratio	Fineness (tex = mg/m)	Specific Tensile Stress (cN/tex)		Ultimate Tensile Strength (MPa)		Elongation at Break (%)	
			1st day	33th month	1st day	33th month	1st day	33th month
1780-I	6.5	15	2.4 ± 0.2	2.4 ± 0.3	28.6 ± 3.1	28.1 ± 3.3	263.1 ± 70.7	78.2 ± 32.0
1781-I	10.0	10	2.6 ± 0.3	2.5 ± 0.2	30.6 ± 3.2	30.1 ± 2.7	112.4 ± 28.4	62.8 ± 15.8
1782-I	10.0	12	3.2 ± 0.2	3.9 ± 0.3	37.9 ± 2.6	46.5 ± 3.2	50.8 ± 8.2	35.9 ± 11.1
1848-II	8.0	5	6.6 ± 1.0	7.1 ± 0.8	79.3 ± 12.0	85.3 ± 9.8	93.7 ± 11.4	57.7 ± 24.8
1783-II	8.0	8	9.3 ± 1.2	10.3 ± 1.5	111.7 ± 14.3	123.2 ± 17.8	46.3 ± 4.5	32.5 ± 10.9
1784-II	8.1	8	17.2 ± 1.5	15.6 ± 1.9	205.9 ± 18.2	187.4 ± 23.1	41.8 ± 5.6	22.1 ± 5.4
1845-III	6.0	10	*	14.2 ± 1.2	*	169.8 ± 14.4	*	34.8 ± 6.0
1846-III	8.0	6	24.2 ± 1.9	18.9 ± 1.2	290.9 ± 22.9	226.9 ± 13.9	37.3 ± 7.9	19.5 ± 3.6
1847-III	8.0	5.4	11.6 ± 1.9	14.4 ± 1.0	139.1 ± 23.5	172.8 ± 12.3	15.9 ± 4.3	15.8 ± 4.5

* data not available.

## Data Availability

The data presented in this study are openly available in the Mendeley repository at DOI: 10.17632/fjr7py9vxv.1.

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
