# Peer review of "Structure–Property Relationship in Melt-Spun Poly(hydroxybutyrate-co-3-hexanoate) Monofilaments"

_polymers, 2022, doi:10.3390/polym14010200_

Round 1
Reviewer 1 Report
polymers-1515308
Title: Structure-property relationship in melt-spun Poly(hydroxybutyrate-co-3-hexanoate) monofilaments
The work is written correctly, and the analysis of the obtained results is interpreted very accurately.
Tables and figures are easy to read.
Conclusions edited correctly.
The following should be addressed:
- Table 1 and the entire text of the manuscript: according to Dispersity in polymer science (IUPAC Recommendation 2009) (Pure Appl. Chem., 2009, Vol. 81, No. 2, pp. 351-353) the term polydispersity index is not used - please correct it
- Please indicate the accuracy of the temperature reading for DSC and TGA analysis or round temperature values to whole numbers
- How can you explain that changes in UTS (Ultimate tensile strength) in some cases caused an increase in UTS and in some a decrease in UTS?
- Please add the statistical analysis of the results (ANOVA).
Major revision
Reviewer 2 Report
Poly(hydroxybutyrate-co-3-hexanoate) (PHBH) is a biodegradable thermoplastic polyester with a potential to be used in textile and medical applications. The authors investigated developing of an upscalable melt-spinning method to produce fine biodegradable PHBH filaments without the use of an ice water bath and offline drawing techniques. The investigations are interesting in biodegradable polymers with medical applications field and could be considered for publication after revision.
- The investigated copolymers have either 11 mol % (grade I) or 6 mol % (grade II, III) of 3-HH. Why these grades are chosen for the investigations ?
- Prior to melt-spinning, the PHBH polymers are dried in a vacuum oven at 90 °C for 12 hours. Why the drying is necessary ? This would be economical disadvantage for practical application ?
-Do the described copolymers have some improved properties as compared with other biodegradable polymers or polymers for medical applications, which are described in literature ?
-Advantages and disadvantages of properties of these polymers should be described in conclusions as compared with other polymers used in similar application fields.
-The authors should demonstrate application of the presented co-polymeric products in textile ord medical applications.
Round 2
Reviewer 1 Report
Manuscript has been significantly improved. I accept it in its current form.
Reviewer 2 Report
If editor and other reviewers agree I recommend the paper for publication after the revision.